# Ochratoxin A Defective *Aspergillus carbonarius* Mutants as Potential Biocontrol Agents

**DOI:** 10.3390/toxins14110745

**Published:** 2022-10-31

**Authors:** Belén Llobregat, Luis González-Candelas, Ana-Rosa Ballester

**Affiliations:** Instituto de Agroquímica y Tecnología de Alimentos (IATA), CSIC, Calle Catedrático Agustín Escardino 7, 46980 Paterna, Spain

**Keywords:** competence, mycotoxin, outcompete, secondary metabolism, VELVET complex, polyketide synthase

## Abstract

*Aspergillus carbonarius* is one of the main species responsible for wine, coffee and cocoa toxin contamination. The main mycotoxin produced by this fungus, ochratoxin A (OTA), is a secondary metabolite categorized as a possible carcinogen because of its significant nephrotoxicity and immunosuppressive effects. A polyketide synthase gene (*otaA*) encodes the first enzyme in the OTA biosynthetic pathway. It is known that the filamentous fungi, growth, development and production of secondary metabolites are interconnected processes governed by global regulatory factors whose encoding genes are generally located outside the gene clusters involved in the biosynthesis of each secondary metabolite, such as the *veA* gene, which forms part of the VELVET complex. Different fungal strains compete for nutrients and space when they infect their hosts, and safer non-mycotoxigenic strains may be able to outcompete mycotoxigenic strains during colonization. To determine the possible utility of biopesticides based on the competitive exclusion of mycotoxigenic strains by non-toxigenic ones, we used *A. carbonarius* Δ*otaA* and Δ*veA* knockout mutants. Our results showed that during both in vitro growth and infection of grapes, non-mycotoxigenic strains could outcompete the wild-type strain. Additionally, the introduction of the non-mycotoxigenic strain led to a drastic decrease in OTA during both in vitro growth and infection of grapes.

## 1. Introduction

Mycotoxins are highly poisonous and carcinogenic secondary metabolites produced by several fungi. *Aspergillus*, *Fusarium* and *Penicillium* are the major filamentous fungi associated with the production of the mycotoxins that are hazardous for both human and animal health [1]. Aflatoxins, deoxynivalenol, ochratoxin A (OTA), fumonisins, T-2 and HT-2 toxins, patulin and zearalenone are considered the most relevant mycotoxins from a food/feed safety point of view [2]. The best strategy to lower mycotoxin levels in the food chain is to prevent the growth of the fungus and, if it has already grown, to prevent it from producing the toxin. Different physical (temperature and humidity management), chemical (use of chemical antifungal agents, electrolyzed oxidizing water treatment, etc.), and biological (use of antagonistic microorganisms, plant extracts) methods have been proposed [3]. One of the most important strategies for managing fungal diseases is to employ antagonistic microorganisms. The four main mechanisms deployed by biocontrol microorganisms are: (i) competing for resources and available space; (ii) producing antibiotics; (iii) inducing resistance; and (iv) direct parasitism. In addition to bacteria and yeast, it is also feasible to use filamentous fungi as biocontrol agents, and fungi that do not produce toxins and can compete with mycotoxigenic fungi are especially interesting. The mechanism of mycotoxin biocontrol is primarily attributed to competitive exclusion, but mycotoxins are sometimes reduced by a larger amount, which can be explained by the displacement of mycotoxin-producing fungi on the crop. Some examples are the uses of aflatoxigenic strains of *Aspergillus flavus*. Different products are commercially available: *A. flavus* AF36 (AF36 Prevail ®) was the first atoxigenic *A. flavus* strain-based product to be registered worldwide for the biological control of aflatoxins in fields in 2003 [4]; *A. flavus* strain NRL18543 is the active ingredient of this product. In 2004, the U.S. Environmental Protection Agency also registered Afla-Guard ®, which contains *A. flavus* NRRL21882 as the active agent [5], and it completely lacks both aflatoxin and cyclopiazonic acid [6]. *A. flavus* Og0222, the biocontrol ingredient of Aflasafe^TM^, was developed for African countries and is commercially available for applications in maize and groundnuts, among others. Atoxigenic *A. flavus* strain MUCL5491 (biocontrol product named AF-X1) has also been approved for use in Italy to control the aflatoxin contamination of maize [7]. These atoxigenic *A. flavus*-based biocontrol products have been developed worldwide and share the characteristic of using native *A. flavus* strains that have been selected for the region where they will be employed. These particular strains have been carefully chosen because they do not produce aflatoxins and/or cyclopiazonic acid [6,8]. However, whether they produce other mycotoxins or harmful secondary metabolites is unknown. In this regard, we should consider the impact that global climatic changes can have on environmental factors, such as water, temperature and carbon dioxide, which could result in the production of new mycotoxins and/or higher levels of already known mycotoxins and, therefore, food safety and security concerns could increase [9,10].

OTA is a toxic secondary metabolite produced by the fungi of the *Aspergillus* and *Penicillium* genera, among others. Exposure to a low concentration of this mycotoxin may lead to neurotoxicity, hepatotoxicity, immunotoxicity and carcinogenicity. Therefore, OTA has been defined as Group 2B (possible carcinogenic to humans) by the International Agency for Research on Cancer (IARC). Comparative genomics has helped us to identify the genes involved in OTA biosynthesis [11,12,13,14]. The role of different genes has been proved: *otaA*, previously designated as *AcOTApks* [15], codes for a polyketide synthase [15,16]; *otaB-AcOTAnrps* codes for a non-ribosomal peptide synthase [17]; *otaD-AcOTAhal* encodes a halogenase [18]. A P450 cytochrome oxidase (*otaC-AcOTAP450*) [12], a bZIP transcription factor (*otaR1-AcOTAbZIP*) [19], and a cyclase *otaY-AcOTASnoaL* gene [20] also form part of the OTA biosynthetic cluster. In addition to the specific genes of the OTA biosynthetic pathway, different studies have shown the involvement of global transcription factors in the regulation of secondary metabolites biosynthesis in *A. carbonarius*, including genes *veA* and *laeA* [16,21]. Disruption of either of these two genes significantly reduces OTA production during in vitro and in vivo growth. 

Recent works have established that climate change can modify the OTA levels produced by *A. carbonarius* [9,10,22]. Thus, ensuring that the selected strain does not produce mycotoxins regardless of environmental changes in temperature or humidity is crucial. The working hypothesis of this study is that the application of non-ochratoxigenic strains might lower mycotoxin levels in food and feed. In this work, we focused on two genes for the generation of non-OTA-producing *A. carbonarius* strains: *otaA* and *veA*. While the deletion of the *otaA* gene should affect only OTA production, the deletion of *veA* could alter the production of other secondary metabolites [23,24] to pave the way to develop safer non-mycotoxigenic strains. We investigated the putative biocontrol activity of these two mutants and determined the lowering of mycotoxin levels as a consequence of the displacement of the wild-type (wt) strain during the co-cultivation of wt and mutant strains in culture medium and during grape berry infection.

## 2. Results

### 2.1. Construction and Characterization of Knockout Mutants 

In order to study whether a non-mycotoxigenic strain could outcompete a mycotoxigenic strain, we constructed two different knockout mutants in the OTA-producing *A. carbonarius* ITEM 5010 background [17]: (i) one based on the deletion of the *otaA* gene (the mutant denoted as Δ*pks*), the first gene in the OTA biosynthesis; (ii) a second one based on the *veA* gene (the mutant denoted as Δ*veA*), and a global regulator of secondary metabolism. The followed gene replacement strategy is shown in Appendix A. Gene replacement plasmids pRFHU2-pks and pRFHU2-veA were obtained by the USER-Friendly cloning strategy [25]. The co-cultivation of the *Agrobacterium tumefaciens* cells carrying the plasmid with the conidia of *A. carbonarius* led to hygromycin-resistant colonies appearing approximately 3 days after being transferred to selective PDA plates. PCR amplification was used to check the correct insertion of the T-DNA containing the hygromycin-resistant marker that replaced the gene of interest (Appendix A). To select the knockout mutants with no further T-DNA integrations, the copy number of the integrated T-DNA was determined by quantitative real-time PCR (qPCR) (Appendix A). Three knockout mutants and two ectopic mutants for each construction were selected for further analyses.

The phenotypic traits of the wt strain, and the ectopic and knockout mutants regarding growth, conidiation and OTA production, were recorded on PDA plates after 5 days post inoculation (dpi) of growth (Figure 1). The secondary metabolism was impaired in the Δ*veA* mutants. It resulted in a brownish pigment observed at the back of the colony compared to the wt strain and the Δ*pks* mutants (Figure 1A). Non-statistically significant differences for the growth area were detected among the wt as well as the ectopic and Δ*pks* mutants (Figure 1B). However, the growth of the Δ*veA* mutants was significantly lower than that of the wt and ectopic mutants (Figure 1E). The conidiation of the three Δ*pks* mutants and the three Δ*veA* mutants was lower than for the wt strain (Figure 1C,F). The Δ*veA* mutants displayed a marked reduction in conidiation, approximately 50 % compared to the wt strain. The OTA production analysis showed that none of the Δ*pks* and Δ*veA* knockout mutants was able to produce the mycotoxin under the assayed conditions (Figure 1D,G). These results were confirmed by HPLC-MS (data not shown).

### 2.2. Different Stresses Did Not Affect the Growth of the Knockout Mutants

In order to investigate the sensitivity of mutants Δ*pks* and Δ*veA* to several stresses, the potato dextrose broth (PDB) supplemented with several concentrations of different stressor compounds was tested (Figure 2 and Figure 3). The assay included high osmolarity, tested with sorbitol, osmotic stress induced by sodium salt (NaCl) and compounds that affect the cell wall and cell membranes, such as calcofluor white (CFW) and sodium dodecyl sulfate (SDS), respectively. Oxidative stress was tested with different oxygen peroxide (H_2_O_2_) concentrations. The wt strain growth was not affected by increasing the H_2_O_2_, CFW or sorbitol concentrations, but it diminished at pH 7.5 compared to the more acidic pHs of 3.0, 4.5 and 6.0. Moreover, its growth was severely affected by adding high NaCl and SDS concentrations. Compared to the wt, the Δ*pks* mutants did not show significant differences for any of the tested compounds or pHs (Figure 2).

No significant differences were observed for the Δ*veA* mutants compared to the wt when H_2_O_2_, SDS or sorbitol were added to culture media (Figure 3). However, the addition of CFW at 1125 and 2250 µg/mL reduced the growth of both Δ*veA* mutants. Although the addition of 1125 mM NaCl modified their growth, the overall curve profile was the same for the three strains. The pH of the medium affected the growth of the Δ*veA* knockout mutants, which was higher at pH 3.0 and lower at 7.5 compared to the growth of the wt.

### 2.3. Competitiveness of Mutants Δpks and ΔveA during In Vitro Growth

In order to assess the ability of mutants Δ*pks* and Δ*veA* to compete with the wt strain, we co-inoculated the wt and the mutants at different ratios (10 wt:1 Δ, 1 wt:1 Δ, 1 wt:10 Δ) on a 96-well microplate. The control wells were inoculated with either the wt or knockout mutants, and they were cultured under the same conditions. The percentage of each strain at the beginning of the experiment was verified by the colony-counting method, and by inoculating the PDA plates supplemented, or not, with antibiotic (Appendix A). Only the knockout strains were able to develop on the antibiotic-supplemented PDA plates. Each well was collected independently after 7 days of incubation at 28 °C for further analyses. The percentage of each strain under competition was calculated by the colony-counting (Appendix A) and qPCR (Figure 4A,B) methods. Based on the qPCR data, when the mutants and the wt strain were inoculated at the same ratio (1 wt: 1 Δ), mutants were capable of displacing the wt strain at 14.5% wt: 85.5% Δ*pks*, and 25.6% wt: 74.4% Δ*veA* (Figure 4A,B). Furthermore, mutants were able to displace the wt even under the most unfavorable condition (10 wt:1 Δ ratio). The Δ*pks* mutant succeeded in imposing itself on the wt more intensely than the Δ*veA* mutant.

Regarding OTA production, the wt strain synthesized OTA, but neither the Δ*pks* (Figure 4C) nor the Δ*veA* (Figure 4D) mutant produced OTA under the assayed conditions. OTA was not even detected in the 1 wt: 10 Δ*pks* mixture culture. Under the most unfavorable growth condition (10 wt: 1 Δ), OTA production lowered from the expected value of 90.9 % to 62.5 % and 68.0 % when the wt was co-inoculated with the Δ*pks* mutant or the Δ*veA* mutant, respectively.

### 2.4. Competitiveness of the Δpks and ΔveA Mutants during Grape Berry Infection

The results of the co-inoculation of the wild-type strain and the mutants in grape berries suggested that both mutants were able to lower the amount of OTA produced by the wt strain (Figure 5). Both mutants Δ*pks* and Δ*veA* were capable of diminishing OTA production from the expected value of 50.0% to 21.1% and 10.1%, respectively, when the wt strain and mutants were equally inoculated (1 wt:1 Δ). Moreover, the presence of the Δ*pks* mutant reduced OTA production from an expected value of 90.9% to 30.7% under the most adverse condition (10 wt:1 Δ).

## 3. Discussion

The present study investigated the potential application of non-toxigenic knockout mutants of *A. carbonarius* for reducing the OTA produced by the ochratoxigenic *A. carbonarius* ITEM 5010 wt strains. Different commercial products are already available for the application of non-toxigenic *A. flavus* strains [6,7,26], which supports the notion that competitive exclusion occurs between toxigenic and non-toxigenic strains [27]. Afla-Guard^®^ and AF36 are two commercial biological control products based on non-aflatoxigenic *A. flavus* strains, which have been approved by the U.S. Environmental Protection Agency for the biological control of *A. flavus* and aflatoxin contamination in peanut, corn and cottonseed with considerable success [28]. The addition of atoxigenic *A. flavus* strain A2085 (AF-X1^TM^) in artificially inoculated maize ears very significantly lowers aflatoxin concentrations [7]. Despite the good results obtained in the control of aflatoxins produced by *A. flavus*, there are no commercial products available to minimize the OTA produced by *A. carbonarius*. Numerous research works on this fungus have focused on natural strains that do not produce OTA and on deciphering the genetic causes of non-mycotoxin production. A common putative cluster of genes involved in OTA biosynthesis has been described in *A. carbonarius* [11,12,13], and OTA production is regulated by different biosynthetic genes, such as genes *otaA-AcOTApks* and *otaB-AcOTAnrps*, among others [15,16,17]. Other general regulatory genes, such as *veA* and *laeA*, impact mycotoxin production [16,21]. Previous research has demonstrated that co-inoculation of an OTA-producing *A. carbonarius* strain with a non-OTA-producing *A. carbonarius* strain, which harbors a Y728H mutation in the *otaA* gene, can lower OTA levels [29]. If, as suggested by the authors, failed OTA production in this strain is due to a single point mutation, then there is a chance that this mutation can be reversed and lead back to an OTA-producer strain. It is well-known that climate change can affect the production of mycotoxins, and an increase in CO_2_ on a grape-based matrix often stimulates the development of *A. carbonarius* and OTA production [22]. With the present climate change scenario, it is critical to ensure that mycotoxins are never synthesized despite changing environmental conditions. 

In this work, we specifically wished to determine whether the use of non ochratoxigenic strains obtained by deleting either the *otaA* genes or *veA* genes could reduce OTA contamination in foods when competing with their parental mycotoxigenic strain. The deletion of the *A. carbonarius otaA* gene led to reduced conidiation (Figure 1C) and, as expected, no OTA production was detected during either *in vitro* growth on rich medium (Figure 1D) or grape berry infection (Figure 5). Other authors have also reported similar results about Δ*pks* mutants’ lack of OTA production [15,16]. Nevertheless, when growing on a minimal medium at 25 °C, Gallo et al. [15] did not observe any phenotypic differences between the wt *A. carbonarius* ITEM 5010 and the Δ*AcOTApks* strain. When grown on yeast extract sucrose medium at pH 4.0 and 28 °C, *A. carbonarius* isolate NRRL 368 and the Ac*pks*-knockout mutant showed comparable growth, sporulation and conidial germination patterns [16]. The different strains utilized in these research works, the incubation temperature and the culture medium are all variables that may influence how the fungus behaves. However, none of the mutants produced OTA, which confirms that this gene is essential for the synthesis of this mycotoxin.

The heterodimeric VELVET complex VelB/VeA/LaeA is a fungus-specific protein complex that controls development, secondary metabolism and pathogenicity. In *A. carbonarius*, *VeA* gene deletion resulted in decreased conidiation, which indicates that this gene is essential for the strain’s ability to produce conidia (Figure 1) [21]. In contrast, VeA acts as a negative regulator of asexual development in *Aspergillus cristatus* and *Aspergillus nidulans*, among others [30,31]. Although it would appear that this gene plays a species-specific role in the regulation of conidia formation, its role in the production of several mycotoxins is clear: although VeA generally plays a positive role in the regulation of secondary metabolite production, some secondary metabolites are negatively regulated by VeA [24,32]. In line with that previously shown by Crespo-Sempere et al. [21], OTA production was affected by the deletion of the *veA* gene in *A. carbonarius* (Figure 1). This gene also positively regulates OTA biosynthesis in *Aspergillus niger* and *Aspergillus ochraceus* [33,34], and it plays a role in the production of aflatoxins in *Aspergillus flavus* [35] and fumonisin by *Fusarium verticillioides* [36]. Light, pH, and osmotic and oxidative stresses are a few examples of environmental stressors that VeA responds to. For instance, in the dark, the OTA production in the AcΔ*veA* mutant decreased to a greater extent compared to the wt *A. carbonarius* when both were grown under light conditions. Nevertheless, conidia production similarly diminished in both situations compared to the wt [21]. *A. carbonarius* exhibits different behavior depending on the pH of the environment. At pH 7.0, it produces less OTA than at pH 4.0 [37]. Our results showed that the growth of both the Δ*pks* mutant and the wt was marginally lower under the neutral pH 7.5 conditions than under more acidic conditions (Figure 2). However, the Δ*veA* mutant developed more quickly under the most acidic situation that we examined but less rapidly under neutral conditions (Figure 3). Previous studies have shown that the disruption of the *veA* gene reduces tolerance to oxidative stress in *A. flavus* and *A. niger* [34,38], but it has no effect on *A. fumigatus* [39]. Our study demonstrated that loss of genes *veA* and *otaA* in *A. carbonarius* did not change the behavior of the mutants in a liquid medium with hydrogen peroxide employed as an oxidant inducer. This finding suggests that *veA* performs no function in preventing oxidative stress. We investigated how osmotic stressors, such as NaCl and sorbitol, affect growth in *A. carbonarius,* and whether genes *otaA* and *veA* could impact this. Our results revealed that osmotic stress caused by high NaCl concentrations leads to diminished fungal growth. However, the growth in both mutants was unaffected compared to the wt, which is similar to that described in *A. flavus,* where the response to osmotic is barely affected by VeA [38].

The parallel differentiation and quantification of *A. carbonarius* toxigenic and non-toxigenic strains in mixed cultures were completed by qPCR following a similar strategy to that used to discriminate non aflatoxigenic biocontrol strains and aflatoxigenic strains of *A. flavus* by droplet digital PCR and qPCR [40,41]. The qPCR results (Figure 4) were the equivalent to those obtained by the traditional plate-counting approach using PDA supplemented, or not, with Hygromicin B as a selective marker for mutants (Appendix A). Even under the most unfavorable initial inoculation conditions (10 wt:1 Δ), both mutants Δ*pks* and Δ*veA* were able to displace the wt strain. A low percentage of the non-toxigenic *A. carbonarius* was able to significantly lower the OTA level in both culture medium (Figure 4) and infected grapes (Figure 5). The overall results herein described indicate that the knockout mutants reduced OTA production to a greater extent than growth. It is worth noting that the co-inoculation of grapes with a low percentage of the Δ*pks* mutant resulted in a more pronounced OTA reduction than with the Δ*veA* mutant. This suggests that the Δ*veA* mutant displays less fitness than the Δ*pks* mutant during grape infection in the presence of the wt strain. This differential fitness could be related to the role of the missing proteins. As previously mentioned, OtaA is a polyketide synthase that contributes specifically to OTA biosynthesis, while protein VeA is involved in global regulation and affects several functions, such as sexual development, secondary metabolism, and even pathogenicity. As VeA is a global regulator of secondary metabolism, it could be advantageous to utilize Δ*veA* mutants if the goal of using biocontrol agents is to eliminate the possibility of any mycotoxin being present, and not only OTA. Further experiments should be run to test this hypothesis.

The preliminary results of this study using these mutants as biocontrol agents are promising for the control of OTA. Similarly to what happens with *A. flavus* [7], it could be feasible to employ non ochratoxigenic mutants to outcompete toxigenic *A. carbonarius* strains to reduce OTA contamination in food and feed.

## 4. Materials and Methods

### 4.1. Fungal Strains and Culture Conditions

*Aspergillus carbonarius* ITEM 5010 was used as the wt strain to construct knockout mutants for genes *otaA* and *veA*. All the strains were grown on potato dextrose agar (PDA, Difco-BD Diagnostics, Sparks, MD, USA) or PDB, with or without the corresponding antibiotic. Cultures were incubated at 28 °C for 7–14 days. Conidia were scraped off agar with a sterile spatula, suspended in distilled water and titrated by a hemacytometer. Plasmid vectors were cloned and propagated in *Escherichia coli* DH5α grown in Luria–Bertani medium (LB; bacto tryptone 10 g, yeast extract 5 g, NaCl 5 g, agar 15 g, per liter) supplemented with 25 µg/mL of kanamycin at 37 °C. *Agrobacterium tumefaciens* AGL-1 was grown in the LB medium supplemented with 20 µg/mL of rifampicin, 100 µg/mL of kanamycin and 75 µg/mL of carbenicillin at 28 °C.

### 4.2. Construction and Verification of the A. carbonarius Knockout Mutants

In order to construct the *otaA* and *veA* gene replacement plasmids, the 5’ and 3’ flanking regions (≈1.5 kb) were amplified from *A. carbonarius* ITEM 5010 genomic DNA and cloned into plasmid vector pRF-HU2 [25]. All the primers pairs were designed by the Primer3 software [42]. The amplification of 10 ng of genomic DNA was performed using High-Taq DNA polymerase (Bioron GmbH, Ludwigshafen, Germany) according to the manufacturer’s instructions and using primers pairs: veA-VA-O1/veA-VA-O2 and OTApks_O1/OTApks_O2 for the upstream regions; veA-VA-A3/veA-VA-A4 and OTApks_A3/OTApks_A4 for the downstream regions (Appendix A). The cycling conditions consisted of: 94 °C for 3 min, 35 cycles of 94 °C for 15 s, 58 °C for 20 s and 72 °C for 1 min 30 s and 72 °C for 10 min.

Binary vector pRF-HU2 was designed to be used with the USER (Uracil-Specific Excision Reagent) friendly cloning technique (New England Biolabs, Ipswich, MA, USA), as previously described [43]. First to obtain plasmids pRFHU2-VEA and pRFHU2-PKS (Appendix A), the amplified upstream and downstream fragment regions were mixed with digested vector pRF-HU2 (ratio of 30:30:120 ng) and the USER enzyme mix, and then, they were incubated according to the manufacturer’s conditions. An aliquot of the mixture was directly used for the transformation of the *E. coli* DH5α chemically-competent cells. Kanamycin-resistant transformants were screened by PCR using primers pairs RF-5/RF-2 and RF-1/RF-6 (Appendix A). Proper fusion was confirmed by DNA sequencing.

The transformation of *A. tumefaciens* was completed by introducing both plasmids independently in electrocompetent *A. tumefaciens* AGL-1 cells with a Gene Pulser apparatus (Bio-Rad, Richmond, CA, USA) [44]. Then, the final *A. carbonarius* transformation was completed as previously described [44] by mixing equal volumes of a conidial suspension of *A. carbonarius* (10^4^, 10^5^ and 10^6^ conidia/mL) and IMAS-induced *A. tumefaciens* cultures, and spreading them onto paper filter layered on agar plates containing IMAS. After co-cultivation at 24 °C for 40 h, membranes were transferred to PDA plates containing hygromycin B (HygB, 100 µg/mL, InvivoGen, San Diego, CA, USA) to select fungal transformants and cefotaxime (200 µg/mL, Sigma-Aldrich, St. Louis, MO, USA) as an inhibitory growth agent of *A. tumefaciens* cells. Finally, *A. carbonarius* ITEM 5010 hygromycin-resistant colonies appeared after 3 to 4 days of incubation at 28 °C, which allowed monosporic cultures to be subsequently obtained.

Genomic DNA extraction was completed as formerly described [43]. The PCR analysis of the transformants was used to confirm the disruption of genes *veA* and *otaA*: screening the deletion of the *veA* gene (primers pairs: veA-VI/veA-VJ), the deletion of the *otaA* gene (primers pairs: OTApks-F3/OTApks-R4), and the insertion of selection marker HygB (primers pairs: HMBF1/HMBR1) (Appendix A). Finally, to check the number of T–DNA copies integrated into the genome of transformants, qPCR was carried out as previously described [44]. The primers pairs used to determine the T-DNA copy number of genes *veA* and *otaA* were designed in the upstream region of the genes with primers pairs veA-VG/veA-VH, and OTApks-R4/OTApks-F8, respectively. The gene selected as a reference was the non-ribosomal peptide synthetase (*nrps*) gene (ID: 132610), which was detected with primer pairs AcNRP_F/AcNRP_R (Appendix A) [21]. Then, qPCR reactions were performed as formerly reported [44] in a LightCycler 480 Instrument (Roche Diagnostics, Mannheim, Germany) equipped with the LightCycler SW 1.5 software and by calculating the number of T–DNA copies integrated into the genome of each transformant according to Pfaffl [45].

### 4.3. Characterization of the Knockout Mutants: Mycelial Growth, Conidiation and Growth under Stress Conditions

For the phenotypical characterization, 5 µL of conidial suspension (1 × 10^5^ conidia/mL) was placed in the center of PDA plates and incubated for up to 5 days at 28 °C. Plates were scanned daily, and the area of growth was analyzed with ImageJ 1.53q (Wayne Rasband, National Institutes of Health, Bethesda, MD, USA). To determine conidia production, three plugs (5 mm diameter) were collected from the center, middle and border of the colony, and 500 µL of methanol was added to the three plugs. The mixture was shaken in an Omni bead Ruptor 24 (Omni International Inc., Kennesaw, GE, USA) for 1 min and at speed 4. Conidia were counted with a hemacytometer and methanol extracts were stored at −80 °C to determine OTA production.

Growth profiles were performed on the 96-well PDB plates supplemented with different concentrations of compounds H_2_O_2_, SDS, CFW, NaCl and sorbitol, and at distinct pHs (3.0, 4.5, 6.0, and 7.5). Spores were collected after 7 days of growth from the pointed-centrally inoculated PDA plates, and concentration was adjusted using a hemocytometer. To determine growth profiles, the 96-well PDB plates containing 100 µL of medium were inoculated in triplicate at a final concentration of 1 × 10^5^ conidia/mL, and they were incubated at 24 °C for up to 7 days. Absorbance at 600 nm was measured automatically at 2 hourly intervals with a FLUOstar Omega (BMG Labtech). Growth curve analyses were performed in R 4.1.1. The area under the curve (AUC) used to describe the growth of the fungi under different stress conditions was calculated at 7 dpi for each replicate based on the spline fit model included in the ‘grofit’ package [46].

### 4.4. In Vitro Competition Assays

Under controlled laboratory conditions, the competition assays involving the ochratoxigenic *A. carbonarius* ITEM 5010 strain and the non-mycotoxigenic mutants were performed on 96-well PDB plates. One knockout mutant for each gene (Δ*veA*10b and Δ*pks*8c) was selected. Spore suspension concentrations were adjusted to 1 × 10^5^ conidia/mL. Five 10:0, 10:1, 1:1, 1:10, and 0:10 mix ratios (the wt strain vs. Δ) were used to inoculate PDB on 96-wells plates. Cultures were grown for 7 days at 28 °C in the dark. Each competition ratio was performed in 15 wells and was repeated twice on different days. At the end of the competition assays, five wells were used to estimate the growth percentage of each competing strain. Then plates were stored at −20 °C for further analyses.

Each strain’s percentage of growth was calculated by two different approaches: (i) diluting the content of the well with 1 mL of water, counting the conidia concentration in at least five wells and plating 100 µL of 2 × 10^3^ conidia/mL onto PDA plates and the PDA plates containing 100 µg/mL of hygromycin. After 24–48 h at 28 °C, incipient colonies were counted, and the percentages of the wt and knockout mutants were determined for each ratio. Only the knockout mutants were able to grow on the DA plates containing hygromycin; (ii) quantifying fitness by qPCR. At least three wells per ratio were used for DNA extraction. The DNAs of *A. carbonarius* ITEM5010 and the knockout mutants were used as a proxy to determine each strain’s growth percentage by employing specific primers: (i) AcveA-VI/AcveA-VJ, and OTApks-F5/OTApks-R6 to detect gene *veA* and gene *otaA*, respectively, in the wt strain; (ii) HPH3F/HPH4R for the hygromycin-resistant marker in the knockout mutants; (iii) AcNRP_F/AcNRP_R to detect the reference gene [21]. A qPCR analysis was performed as previously described.

To determine whether non-mycotoxigenic mutants could inhibit ochratoxin A production by the wt strain *A. carbonarius* ITEM 5010, OTA was independently extracted from at least three co-inoculated PDB wells in 2 mL tubes containing three 2.7 mm steel beads with 500 µL of methanol with the help of an Omni Bead Ruptor 24 (Omni International, Inc., Kennesaw, GE, USA). Samples were centrifuged and filtered, and methanol extracts were stored at −80 °C until further analyses.

### 4.5. Artificial Inoculation of Grape Berries

For each strain, nine grape berries were surface-sterilized with 2% sodium hypochlorite for 5 min, rinsed with tap water and air-dried. Conidia, collected by scraping the surface of 7-day-old colonies grown on PDA plates, were resuspended in water and adjusted to 1 × 10^5^ conidia/mL. Five (wt strain vs. deletant) 10:0, 10:1, 1:1, 1:10 and 0:10 mix ratios were prepared. Aliquots of 5 µL of the conidial suspensions were inoculated on the grape berries that had been previously wounded with a needle at a 5 mm depth. After 10 days at 28 °C in the dark, three biological replicates containing all three grape berries were collected, frozen in liquid nitrogen and ground to a fine powder. Pooled samples were stored at −80 °C until further use. To determine OTA levels, 0.5 g of frozen tissues was mixed with 250 µL of methanol and three steal beads, as previously described, and the methanol extracts were stored at −80 °C to determine OTA production.

### 4.6. OTA Quantification

OTA was detected by a previously described high-pressure chromatography (HPLC) method [44] with minor modifications. Briefly, methanolic extracts were filtered and injected into HPLC. OTA detection and quantification were completed by an HPLC system (ACQUITY Arc Sys Core 1–30 cm, Waters Co., Milford, CT, USA) equipped with a Waters temperature control module, a Waters 2475 fluorescence detector (excitation wavelength of 330 nm and emission wavelength of 460 nm), and a Kinetex biphenyl column (4.6 mm × 150 mm, 5 µm, Phenomenex Inc., Torrance, CA, USA). Twenty microliters of each extract were injected. The mobile phase was acetonitrile: water: acetic acid (57:41:2, *v/v/v*) with isocratic elution for 15 min at a flow rate of 1 mL/min. Working standard solutions were prepared by appropriate diluting the known volumes of the stock solution with methanol, and they were used to obtain calibration curves in the chromatographic system. OTA was expressed as a percentage of the wt strain.

Samples were also analyzed by mass spectrometry with a Waters Acquity UPLC I-Class System (Waters Corporation, Milford, MA, USA) coupled to a Bruker Daltonics QToF-MS mass spectrometer (maXis impact series, resolution ≥ 55,000 FWHM Bruker Daltonics, Bremen, Germany) using ESI for the positive (ESI (+)) ionization mode. UPLC separation was performed in an ACE-Excel C18-PFP column (3.0 mm × 100 mm, 1.7 µm size particle) at a flow rate of 0.3 mL/min. Separation was completed using water with formic acid at 0.01% as the weak mobile phase (A) and methanol with formic acid at 0.01% as the strong mobile phase (B). The gradient started with 20% of B at 0 min, which was maintained for 3 min. Next, the proportion was set at 40% and linearly increased to 80% until minute 22.50. Eluent B was set at 20% and kept constant until minute 30. Nitrogen was used as the desolvation gas with a flux of 9 L/min and as the nebulizing gas with a flux of 2.0 bar. The drying temperature was 200 °C and the column temperature was 40 °C. The voltage source was 4.0 kV for ESI (+). The MS experiment was completed by employing HR-QTOF-MS, applying 24 eV for ESI (+) and using broadband collision-induced dissociation (bbCID). The MS data were acquired within an *m/z* range of 50–1200 Da. 

The external calibrant solution was delivered by a KNAUER Smartline Pump 100 with a pressure sensor (KNAUER, Berlin, Germany). The instrument was externally calibrated before each sequence with 10 mM sodium formate solution. The mixture was prepared by adding 0.5 mL of formic acid and 1.0 mL of 1.0 M sodium hydroxide to an isopropanol/Milli Q water solution (1:1, *v/v*). UPLC-QToF-MS analyses were performed at the Metabolomics Platform of CEBAS-CSIC, Campus Universitario de Espinardo, 30100 Espinardo, Murcia (Spain).

### 4.7. Statistical Analysis

The ANOVA analysis was used to determine if there were significant differences among means. Tukey’s test was carried out to determine if significant (*p* < 0.05) differences occurred between individual treatments (Statpoint).

## Figures and Tables

**Figure 1 toxins-14-00745-f001:**
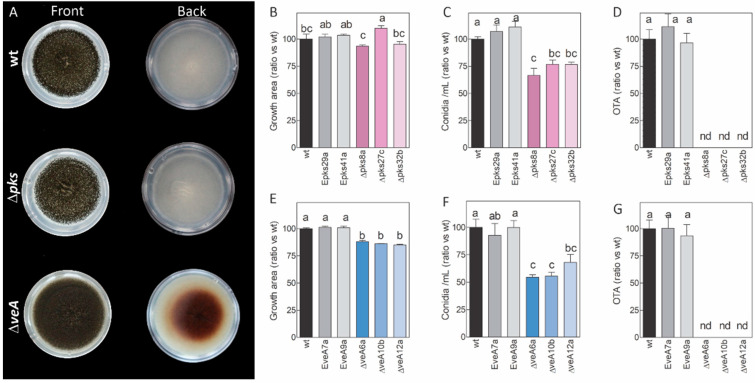
Phenotypic traits of the wild-type strain of *A. carbonarius* ITEM 5010 (denoted as ‘wt’, black bars), two ectopic mutants (denoted as ‘E’, gray bars) and three knockout mutants (color-filled bars) of genes *otaA* (**B**–**D**) and *veA* (**E**–**G**), denoted as Δp*ks* and Δ*veA*, respectively. (**A**) The front and back colony views of the different strains point-inoculated on PDA plates in the dark at 7 days post-inoculation. Growth area (**B**,**E**), conidiation (**C**,**F**) and OTA production (**D**,**G**) were tested on the PDA plates centrally point-inoculated with 5 uL of 1 × 10^5^ conidia/mL. After incubation at 28 °C for 5 days, colonies were scanned to determine the growth area with the ImageJ software. Three plugs were collected from the center, middle and inner parts of the colony for conidia determination and OTA extraction purposes. Values were normalized to those of the wt growing under the same conditions. Error bars represent the standard error of the mean of at least three biological replicates. The bars with different letters in the same panel are statistically different as determined by the one-way ANOVA and Tukey’s test (*p* < 0.05). nd. not detected under the assayed conditions.

**Figure 2 toxins-14-00745-f002:**
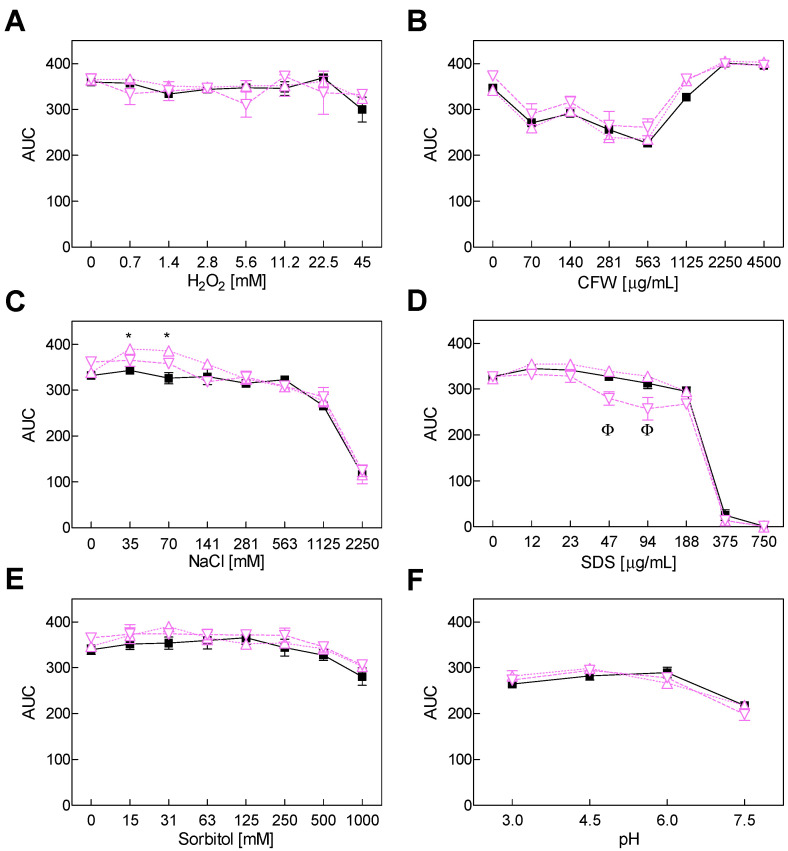
Growth of the wild-type strain *A. carbonarius* ITEM 5010 (■) and two Δ*pks* mutants (△ represents Δ*pks8a*, and ▽ denotes Δ*pks27c*) in the presence of different H_2_O_2_ (**A**), CFW (**B**), NaCl (**C**), SDS (**D**) and sorbitol (**E**) concentrations and at distinct pHs (**F**). Growth was determined as the area under the curve (AUC) after 7 days of incubation at 24 °C. The two-way ANOVA, followed by Tukey’s test (*p* < 0.05), was performed to determine the significant growth of the Δ*pks8a* (denoted as *) and Δ*pks27c* (denoted as Φ) knockout mutants compared to the wild type. Values represent the mean ± standard error of the mean of three biological replicates. The experiment was repeated twice.

**Figure 3 toxins-14-00745-f003:**
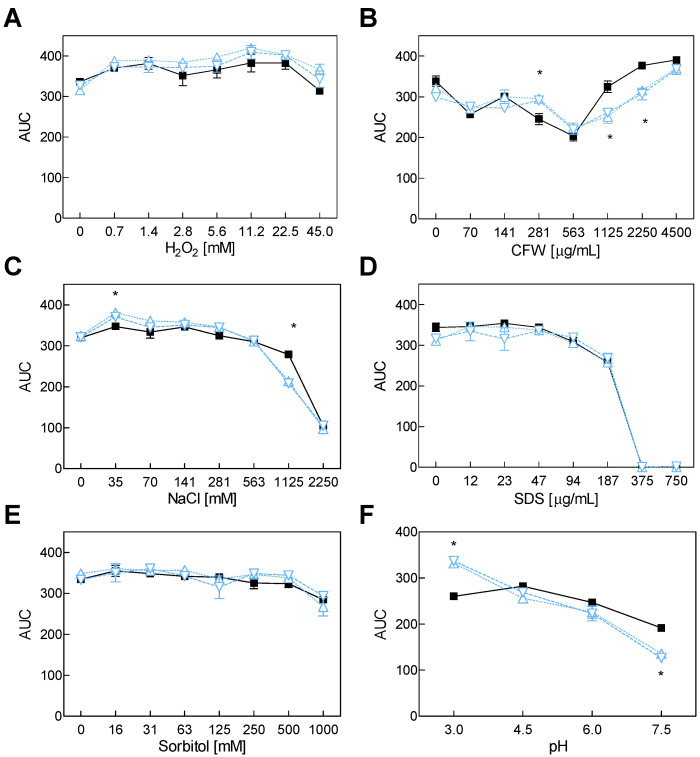
Growth of the wt strain *A. carbonarius* ITEM 5010 (■) and two Δ*veA* (△ represents Δ*veA10b*, and ▽ denotes Δ*veA12a*) in the presence of different H_2_O_2_ (**A**), CFW (**B**), NaCl (**C**), SDS (**D**) and sorbitol (**E**), concentrations and at distinct pHs (**F**). Growth was determined as the area under the curve (AUC) after 7 days of incubation at 24 °C. The two-way ANOVA, followed by Tukey’s test (*p* < 0.05, indicated as *), was performed to establish whether significant differences existed between the knockout mutants and the wild type. Values represent the mean and standard error of the mean of three biological replicates. The experiment was repeated twice.

**Figure 4 toxins-14-00745-f004:**
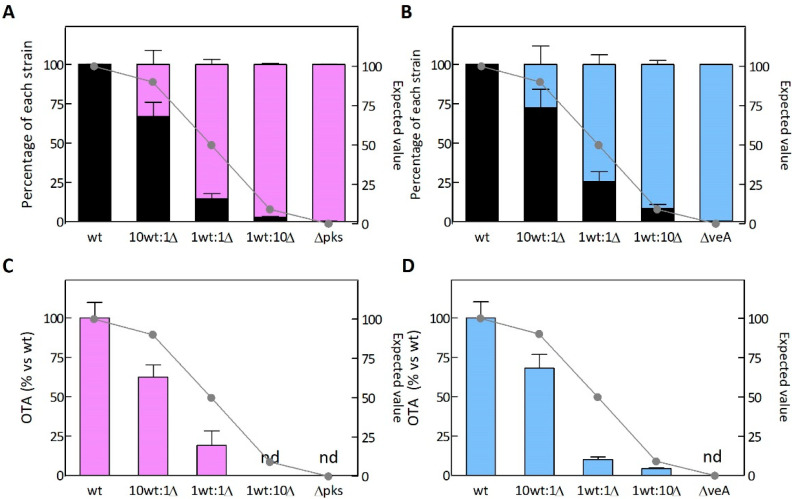
Competitiveness of the Δ*pks* (**A**) and Δ*vea* (**B**) knockout mutants against the mycotoxigenic wild-type (wt) strain *A. carbonarius* ITEM 5010 on potato dextrose broth (PDB). Competition assays were performed at the 1:0, 10:1, 1:1, 1:10 and 0:1 (wt:Δ) ratios. The expected values for the wt strain are indicated as gray dashed lines. The percentage of each strain was estimated by qPCR. Values are the mean of at least three biological replicates and error bars represent the standard error of the mean (SEM). OTA production during the competition assays of (**C**) the wt strain vs. the Δ*pks* mutant, and the (**D**) wt vs. Δ*vea* mutant. Values are normalized based on the OTA production by the wt strain.

**Figure 5 toxins-14-00745-f005:**
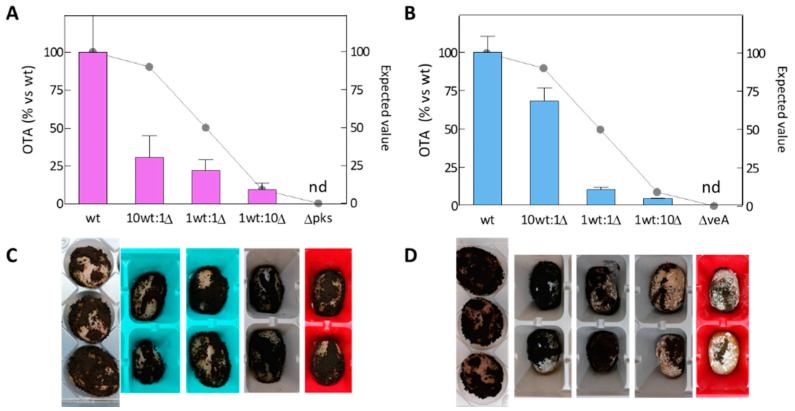
OTA production (**A**,**B**) during the infection of grape berries. Co-inoculation of the mycotoxigenic wil-type (wt) strain *A. carbonarius* ITEM 5010 and the Δ*pks* mutant (**A**,**C**), and the wt and Δ*veA* (**B**,**D**). Competitive assays were performed at the 1:0, 10:1, 1:1, 1:10, and 0:1 ratios (wt:Δ). Grape berries were incubated at 28 °C for up to 10 days. Values are the mean of three biological replicates and error bars represent the standard error of the mean (SEM). Values are normalized based on the production of OTA by the wt strain.

## Data Availability

Not applicable.

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
