# Peer review of "Ochratoxin A Defective Aspergillus carbonarius Mutants as Potential Biocontrol Agents"

_toxins, 2022, doi:10.3390/toxins14110745_

Round 1
Reviewer 1 Report
The article entitled “Ochratoxin A-defective Aspergillus carbonarius mutants as potential biocontrol agents” reports the use of mutants A. carbonarius (without OTA produce capacity) to compete with the wild-type strain. The authors construct knockout mutants for two different genes. It was performed a characterization of mutants growth and compared with wild-type strain growth. The authors made a comparison of stresses response of wild-type and mutants strain and assess the ability of mutants to compete with wild-type strain in vitro and in grape berries. The results show that mutants strains do not produce OTA and have a good capacity to stresses response. The assays about competitiveness showed the use of mutant strains leads to a significant reduction of OTA production. The reduction, both in vitro or in grape berries, was higher than expected (considering the wild-type/mutants ration used on experiments).
This work is very interesting to improve the knowledge about competitive exclusion of mycotoxigenic strains by non-toxigenic ones.
I suggest this manuscript need a minor revision. My critical comments are listed below.
Page 2, Line 47 – Verify if you want to say “atoxigenic strains” instead “aflatoxigenic strains”.
Page 2, Line 51 – What “EPA” is?
Page 10, Line 403 – Change “μl” to “μL”. Check overall text.
Overall text – Check and unifomize the way to presente “conidia mL-1". You have conidia mL-1 and conidia/mL.
Page 11, Line 457 – Verify if the excitation wavelength is 33 nm. Generally, is around 333 nm.

Author Response
Thanks for the reviewer’s comments. In color you can find our replies.
Page 2, Line 47 – Verify if you want to say “atoxigenic strains” instead “aflatoxigenic strains”.
In Line 51, the sentence “Some examples are the uses of aflatoxigenic strains of Aspergillus flavus.” Indicates aflatoxigenic in order to specify that the atoxigenic strains of Aspergillus flavus used as biocontrol strains are deficient in aflatoxin production.
Page 2, Line 51 – What “EPA” is?
Thanks for the comment. We have changed “EPA” by “U.S. Environmental Protection Agency”
Page 10, Line 403 – Change “μl” to “μL”. Check overall text.
We have changed and check the overall text according the reviewer comment.
Overall text – Check and unifomize the way to presente “conidia mL-1". You have conidia mL-1 and conidia/mL.
We have checked the manuscript according the reviewer comment.
Page 11, Line 457 – Verify if the excitation wavelength is 33 nm. Generally, is around 333 nm.
Yes, it was a mistake. The excitation wavelength is 330nm
Reviewer 2 Report
This paper is certainly research worth reading, regardless of whether it is the field the researcher is involved in or not. I only suggest the following:
title: I suggest removing the hyphen and putting a comma, because this way it seems that "A" is connected to the word "defective"
Sincerely
Author Response
Thanks for the reviewer’s comments. In color you can find our replies.
This paper is certainly research worth reading, regardless of whether it is the field the researcher is involved in or not. I only suggest the following:
title: I suggest removing the hyphen and putting a comma, because this way it seems that "A" is connected to the word "defective"
According the reviewer comment, we have eliminated the hyphen.
Reviewer 3 Report
The paper was well written and the work is very significant. I think the work needs moderate English changes.
Author Response
According to the reviewer’s comment, we have submitted the article for correction to a Native English writer. Attached you can find the document confirming the English correction of the manuscript.
